# Recent Advances in Mpox Epidemic: Global Features and Vaccine Prevention Research

**DOI:** 10.3390/vaccines13050466

**Published:** 2025-04-25

**Authors:** Xinling Zhang, Dong-Ang Liu, Yuting Qiu, Ruiyao Hu, Shiyu Chen, Yue Xu, Keyi Chen, Jinghua Yuan, Xiaoping Li

**Affiliations:** Key Laboratory of Artificial Organs and Computational Medicine in Zhejiang Province, Shulan International Medical College, Zhejiang Shuren University, Hangzhou 310015, China; xinlingz_201803@zjsru.edu.cn (X.Z.); 202211002127@stu.zjsru.edu.cn (D.-A.L.); 202211002114@stu.zjsru.edu.cn (Y.Q.); 202211001205@stu.zjsru.edu.cn (R.H.); 202211002102@stu.zjsru.edu.cn (S.C.); 202211003324@stu.zjsru.edu.cn (Y.X.); 202310302201@stu.zjsru.edu.cn (K.C.); 601660@zjsru.edu.cn (J.Y.)

**Keywords:** mpox, zoonotic disease, etiological characteristics, epidemic situation, vaccine prevention

## Abstract

Monkeypox (mpox) is a zoonotic disease (zoonose) caused by the monkeypox virus (MPXV). MPXV, a member of the *Orthopoxviridae* family, is categorized into two clades, Central Africa (I) and West Africa (II), each of which is further subdivided into subclades a and b. Clade I generally causes more serious illness and higher mortality rates, while Clade II results in milder illness. Historically, mpox epidemics were localized to specific regions and countries in Africa. Since 2022, the mpox epidemic, fueled by MPXV Clade IIb, has swiftly spread across various nations and regions, jeopardizing public health and safety. However, starting in 2024, Clade Ib gradually replaced Clade IIb. The notable genetic variation in Clade Ib may provide MPXV with new opportunities to evade the immune system and adapt to hosts. According to the World Health Organization (WHO), from 1 January 2022, to 24 November 2024, there were 117,663 confirmed cases and 2 probable cases, resulting in 263 deaths across 127 Member States in all six WHO regions. As of 9 January 2025, 12 countries outside Africa have reported imported MPXV Clade Ib cases, with secondary cases emerging in the United Kingdom, Germany, and China. Due to the incomplete development of a vaccine specifically for MPXV, the smallpox vaccine remains in use for preventing mpox or for emergency vaccination post-exposure. Therefore, the persistent spread of mpox is still a major concern, requiring greater awareness and vaccination efforts in populations at high risk. This paper aims to summarize the etiological characteristics, epidemic situation, and vaccine prevention efforts for mpox, offering a reference for managing this serious epidemic and ensuring effective scientific prevention and control.

## 1. Introduction

As a new and re-emerging infectious disease, mpox has broken out and spread worldwide, posing a threat to public safety and health. A global outbreak and epidemic of MPXV Clade II occurred in 2022–2023, the first sustained transmission outside Africa since the first human cases of mpox were identified in 1970 [1]. In 2023, Clade Ib cases were diagnosed in the Democratic Republic of the Congo (DRC) and subsequently spread to neighboring countries, leading to an increasingly serious epidemic [2]. On 14 August 2024, the WHO declared mpox a Public Health Emergency of International Concern (PHEIC) for the second time in two years [3]. Under the emergency prevention and control measures of various countries and regions around the world, along with the importation of relevant vaccines, the number of mpox cases worldwide is generally declining. However, some countries and regions are still experiencing outbreaks or epidemics [4] (Figure 1). Because MPXV-specific vaccines are still not fully developed, smallpox vaccines can be used to prevent mpox or for emergency use. These mainly include first-generation vaccines (Dryvax^®^), second-generation vaccines (ACAM1000^®^, ACAM2000^®^), third-generation vaccines (LC16m8^®^), and fourth-generation vaccines (OrthopoxVac^®^) [5]. Notable approved vaccines include JYNNEOS^®^ (MVA-BN), LC16m8^®^, ACAM2000^®^, and OrthopoxVac^®^ [6,7]. OrthopoxVac^®^, a vaccine of the fourth generation, was primarily created and authorized in Russia, utilizing a weakened version of *Orthopoxvirus* to prevent smallpox. However, its use has not been widely discussed due to the lack of clinical research data on mpox [7]. Widely used during the global mpox outbreak, the first three vaccines provide protection by stimulating a cross-immune response that results in the creation of neutralizing antibodies [8]. Additional mRNA and recombinant protein vaccines are under development. It can be observed from the two epidemics that some countries and regions in Africa have not established a public health surveillance and information reporting system, resulting in a lack of case detection, diagnosis, surveillance, and reporting [9]. Moreover, due to objective factors, such as a lack of equipment, weak technology, insufficient medical staff, and a shortage of vaccines, misdiagnoses, missed diagnoses, inaccurate epidemic assessments, and the expansion of the epidemic are common [10,11]. Therefore, the long-term transmission of mpox cannot be ignored. Studies on the etiology, epidemic status, and vaccine-based prevention of mpox are needed to provide references for severe epidemic forms and effective scientific prevention and control.

## 2. The Etiological Characteristics of Mpox

MPXV is an enveloped, double-stranded DNA virus of the genus *Orthopoxvirus* in the family *Poxviridae* [12]. Subtype MPXV is divided into two clades: Central Africa (I) and West Africa (II). Each of these clades can be divided into subclades a and b [9,13] (Table 1). Clade I, found in Central African countries, has higher mortality (up to 11%) [14], while Clade II, prevalent in West African countries, is less severe, with a 6% mortality rate [15]. The global mpox epidemic in 2022 was primarily caused by clade IIb [8], transmitted predominantly through sexual contact, and was more common among men who have sex with men [16,17]. Human-to-human transmission can also occur through close contact with lesions, respiratory droplets, body fluids, and contaminated materials [18]. Human infection with Clade IIb has an incubation period of 7 to 14 days and symptom duration is 14 to 21 days [19]. In 2024, Clade Ib was the reason for the global mpox epidemic [3], which mainly occurred through close family contact, animal contact, and unprotected medical care contact [20,21]. Family or community were the main transmission locations. Human infection with Clade Ib has an incubation period of 5 to 13 days, and the symptoms may last from 4 to 21 days [22]. The main hosts of MPXV in Central, East, and West Africa are rodents (squirrels) and primates (monkeys) [14]. Current exploration of the mode of transmission is mostly limited to close animal-to-human or human-to-human contact [23], and human-to-animal transmission may also occur [24]. At present, polymerase chain reaction (PCR) is the primary method for laboratory diagnosis. MPXV can be detected by PCR in saliva, oropharynx, upper respiratory tract swabs, blood, semen, vaginal fluid, urine, and anal swabs from infected individuals [25,26]. The stability of MPXV enables it to survive for a certain amount of time in vitro, using contaminated surfaces or environments as a medium. It has a half-life of up to 38.75 days in dried blood, 4.57 days in dried semen, and 5.74 days in wastewater [27]. By comparing the different genomes, significant differences between clades can be observed. Some genes may be completely absent or truncated in a given clade, resulting in changes in gene content. The VACV-Cop E5R ortholog was completely absent from Clade I, along with three other genes, including the VACV-Cop A47L and B11R orthologs, as well as K1R, which were truncated. Unlike in Clade II, all four genes (D14L, D15L, D16L, and D17L) were completely deleted, and three genes (D4L, B14L, and B15L) were truncated [28]. This may be associated with the biological properties of different clades through the deletion or truncation of the genes. However, the genetic diversity of Clade Ib is 54% higher than that of Clade I, with the number of nucleotide changes increasing from approximately 96 to around 149. Among these, many mutations are related to genes involved in host immune regulation and viral replication. For example, the B21R (OPG210) gene exhibited three consensus amino acid substitutions (D209N, P722S, and M1741I), which might exist to impair the immune escape ability and host fitness of MPXV. This remarkable genetic variation is driven mostly by APOBEC3-mediated cytosine deamination [29], and it results in a dramatic increase in CT mutations within the MPXV genome [30]. This mutation pattern not only reflects the critical role of APOBEC3 in MPXV evolution but also the rapid and sustained spread of Clade Ib in the population [31]. In addition, the genomes of the MPXV Clade Ib differ significantly from those of other Clade I virus strains that were previously sequenced in the DRC. There was a deletion of approximately 1 kbp in the genome of Clade Ib (relative to the Clade I reference genome NC_003310) that may have interfered with the detection power of Clade I-specific diagnostic PCR-based assays [32]. Therefore, continuous monitoring and in-depth research on the etiology, transmission, pathogenesis, and evolution of, as well as variations in, mpox are urgently needed for the better prevention and diagnosis of MPXV.

## 3. The Prevalence and Characteristics of Mpox

Monkeypox is a zoonose that was previously restricted to outbreaks or epidemics in selected countries in Central and West Africa [9]. In May 2022, MPXV Clade IIb outbreaks emerged in non-endemic regions such as Europe, America, and Asia. By 23 July, the WHO designated the mpox outbreak as a PHEIC. The outbreak spread across six WHO regions, resulting in 92,783 confirmed cases and 171 deaths [33]. Sexual activity played a key role in transmission during this outbreak, particularly among men who have sex with men [34,35,36]. Vertical transmission from mother to child can occur, which can also easily lead to fetal death and abortion [37]. Indirect transmission occurs through contaminated medical devices in hospitals [38]. With the gradual and effective control of the Clade IIb outbreak, despite the WHO’s declaration on 11 May 2023 that mpox outbreaks no longer constituted a PHEIC [39,40], Clade Ib began to gradually replace the effects of Clade IIb. Since the 1970s, the DRC has reported only cases of Clade I mpox, transmitted through human-to-human contact, mostly in small households or via community outbreaks [21]. In April 2023, the DRC reported the first Clade I mpox cases, transmitted through contact in Kango Province, with multiple cases among sex workers. Subsequently, the Kinshasa and South Kivu provinces reported their first Clade I outbreaks due to sexual contact in August and September 2023 [4]. In July 2024, MPXV Clade Ib began spreading from the DRC to neighboring countries, causing sustained community transmission. This led to the WHO once again declaring the mpox outbreak a PHEIC on 14 August 2024 [3]. As of 9 January 2025, 12 non-African countries had reported imported cases of MPXV Clade Ib infection, with Germany and China experiencing multiple secondary cases [41] (Figure 2). Unlike Clade IIb, which is mainly transmitted through sexual contact, the newly emerging Clade Ib is primarily transmitted through close household contact [42]. As a result, among all age groups, infants and children are more likely to be affected by Clade Ib, influenced by factors such as vaccine inequity, low immunity, and malnutrition [13]. At least 500 related deaths have been reported so far [43]. Among the total cases in all 26 provinces of the DRC, children under 15 years of age accounted for 66% of the reported cases and over 82% of deaths [44]. Studies in the DRC have shown that MPXV Clade Ib has a higher transmission probability through female sex workers, higher incidence in females, and a greater risk of mother-to-child vertical transmission. This indicates that Clade Ib has higher transmissibility and may be more virulent [32] (Figure 3).

## 4. Advances in Mpox Vaccines

### 4.1. Second-Generation Smallpox Vaccine-ACAM2000

ACAM2000, a live attenuated vaccine capable of replication, is derived from a monoclonal virus isolate of the first-generation Dryvax vaccine. Typically, one dose is administered percutaneously to actively immunize groups identified as being at high risk of smallpox infection [45]. ACAM2000 has been shown to be non-inferior to the first-generation vaccine (Dryvax) in immunogenicity tests, with a better safety profile and a lower incidence of side effects [46]. Common side effects include localized reactions such as pain, itching, and swelling at the injection site, as well as systemic symptoms like muscle pain, headache, rash, fever, fatigue, tearing, blurred vision, eye pain, and flu-like symptoms [47]. However, serious side effects include severe allergic reactions, encephalitis, encephalomyelitis, encephalopathy, pericarditis, and even death. A study by Decker et al. involving 897,227 individuals who received the ACAM2000 vaccine over an 18-year period reported an incidence of pericarditis at approximately 20 cases per 100,000 recipients [48]. However, according to data from the United States (US) Food and Drug Administration (FDA), the incidence is higher, with approximately 1 case per 175 vaccinated individuals [47]. In addition, the use of ACAM2000 is strictly contraindicated in individuals with impaired cardiac function, pregnant women, newborns, and immunocompromised individuals (those with leukemia, human immunodeficiency virus, etc.) [8,49]. Therefore, it is only suitable for individuals who are not pregnant and have a functioning immune system [50]. The risk of infection is greatly minimized by vaccination, but since no vaccine is 100% effective, contracting mpox is still possible, particularly during outbreaks.

### 4.2. Third-Generation Smallpox Vaccine

#### 4.2.1. JYNNEOS (MVA-BN)

JYNNEOS is a non-replicating live vaccine derived from the Modified Vaccinia Ankara (MVA) virus. In 2019, JYNNEOS was officially licensed in the US [51]. JYNNEOS has been effective in the emergency prevention of mpox in a number of animal models. In August 2022, the FDA granted emergency authorization of the JYNNEOS vaccine for the prevention of mpox, which has proven effective in multiple animal models [52]. The clinical trial evidence suggests that JYNNEOS provides 66% to 85.9% effectiveness when it is administered as two subcutaneous injections (SC) spaced 4 weeks apart [53,54]. Individuals need a booster shot every 2–10 years if they remain exposed to MPXV [55]. In addition, WHO reported that, in eight separate studies that assessed 1222 participants, the seroconversion rates were consistently higher than 98 percent, proving the efficacy of the vaccine [56]. Typically, the JYNNEOS vaccine causes mild side effects, including injection site reactions and transient systemic symptoms (pain, fatigue, itching, chills, and nausea) [57,58]. No adverse effects of myocarditis or pericarditis were observed among 9713 vaccinated individuals enrolled in 19 randomized controlled trials [47]. Based on their studies, Rao et al. concluded that JYNNEOS is safe for immunocompromised people, but with reduced immunogenicity compared to healthy people, after conducting three randomized controlled trials and 15 observational studies of 5775 vaccinated individuals [59]. According to Fontán-Vela et al., the MVA-BN vaccine proved effective in minimizing the risk of MPXV infection in high-risk HIV pre-exposure prophylaxis cases, with reductions of 79% at 75 days and 14% at 79 days after vaccination [60]. Consequently, JYNNEOS offers a safer alternative to ACAM2000. The MVA-BN vaccine is the preferred choice for individuals with weakened immune systems and can be used for initial and post-exposure prophylaxis in pregnant and breastfeeding women [50]. The risk of infection is greatly minimized by vaccination, but since no vaccine is 100% effective, contracting mpox is still possible, particularly during outbreaks.

#### 4.2.2. LC16m8

The LC16m8 vaccine is a live, minimally replicating, attenuated vaccine derived from the Lister (Elstree) strain. It is administered as a single-dose scratch vaccination using a bifurcated needle. LC16m8 was developed in 1975 by the Kaketsuken Research Institute in Japan and was approved for mpox prevention in Japan in August 2022 [61]. Clinical studies on the immunogenicity of LC16m8 have demonstrated the seroconversion of neutralizing antibodies against MPXV by day 28 post-vaccination [62,63]. The vaccine has a favorable safety profile, with side effects being mild and transient. These include injection site reactions and mild systemic symptoms such as swollen lymph nodes, fatigue, fever, rash, redness, and swelling at the injection site [8,58]. According to a WHO report dated 23 August 2024, research on the efficacy of LC16m8 is ongoing. The inferred clinical efficacy of LC16m8 for smallpox and mpox derives from indirect experiments, such as studies on its protective effects in various animal models. LC16m8 vaccination has provided protection to mice, rabbits, and monkeys against deadly MPXV assaults [56]. Although MVA-BN and LC16m8 have been authorized for post-exposure vaccination in children in the US and Japan, vaccination should only be administered when the benefits clearly outweigh the potential risks, particularly in vulnerable populations [50]. The risk of infection is greatly minimized by vaccination, but since no vaccine is 100% effective, contracting mpox is still possible, particularly during outbreaks.

### 4.3. Recent Advances in Vaccine Development

Currently, the global approaches to developing mpox vaccines include live attenuated vaccines, mRNA vaccines, recombinant protein vaccines, and fourth-generation vaccines. Significant progress has been made in mRNA and recombinant protein research. mRNA vaccines can be quickly developed and designed using viral gene sequences. When the virus mutates, companies can swiftly modify the vaccine formula to avert immune escape. Upon successful development, the production process is easy to manage and can be scaled up to satisfy the demand for mass vaccination [64]. The study by Mucker et al. was the first to evaluate the effectiveness of an mRNA vaccine (mRNA-1769) and MVA in preventing mpox in a non-human primate model [65]. Overall, the results showed that the mRNA-1769 vaccine induced fewer lesions and lower viral replication, in addition to stronger, more active viral neutralizing and functional antibodies that improved the ability to control viral replication and alleviate symptoms of disease. Just like BNT166, an mRNA vaccine from BioNTech, showed high immunogenicity, good protection in preclinical models, and 100% protection in mouse and rhesus monkeys against MPXV and related orthopox viruses [66].

Sometime soon, Chinese homegrown mRNA vaccines will enter clinical trials. The Chinese biological vaccine research and development team is leading in this field. By encoding the A35R and M1R proteins of MPXV, they have developed three mRNA vaccines (VGPox 1, VGPox 2, and VGPox 3) targeting MPXV [67]. All three vaccines rapidly elicited antibodies against A35R in mouse studies, and two mRNA vaccine (VGPox 1, VGPox 2) generated anti-A35R and anti-M1R IgG antibodies very rapidly and well, with strong virus neutralization [67]. Additionally, researchers at the Seventh Affiliated Hospital of Sun Yat-sen University developed a multivalent mpox mRNA vaccine containing four immunogenic targets from mpox Clade II isolates, such as A27, A33, B5, and L1, in 2003. The team encapsulated the mRNAs for these antigens in lipid nanoparticles (LNPs). The mRNA-LNP vaccine with four antigens can effectively trigger a humoral immune response at a high dose (5 μg). Even at a low dose of 0.5 μg, it can ensure 100% survival of mice [64]. In addition, researchers at the University of Science and Technology of China created the MPXV-1103 tetravalent vaccine, targeting the B6, A35, A29, and M1 MPXV proteins at an mRNA immunogenicity equivalent to that of conventional smallpox vaccines ACAM2000 and MVA and inducing strong humoral immunity and T cell responses against MPXV [68,69]. Notably, MPXV-1103 generated high levels of neutralizing antibodies even at a low dose of 1 μg, making it a highly efficient and safe vaccine candidate. It is expected to become a critical tool in preventing mpox infections [69]. In another innovative approach using antigenic structure-guided multi-epitope chimerism, the Institute of Microbiology of the Chinese Academy of Sciences designed a ‘two-in-one’ MPXV recombinant protein vaccine called DAM. DAM provides comprehensive protection against MPXV infectious virus particles with a single immunogen, and its neutralizing power is 28 times higher than that of standard live attenuated vaccines [70]. Overall, global research on mpox vaccines and treatments is advancing rapidly. The ongoing mutations of MPXV complicate vaccine development, underscoring the need for a deeper understanding of its pathogenicity and mutation potential.

## 5. Discussion

The usability of MPXV vaccines is determined using the balance between their immunogenicity and safety, with effectiveness and side effects serving as key evaluation criteria. ACAM2000, derived from a live attenuated cowpox virus, provides significant cross-protection against both smallpox and mpox, demonstrating high immunogenicity [71]. However, it carries a higher risk of side effects. Uncontrolled viral replication may lead to accidental or self-inoculation, resulting in conditions such as cowpox eczema and progressive cowpox [47,72]. In contrast, JYNNEOS does not pose this risk. It utilizes a non-replicating Modified Vaccinia Ankara (MVA) virus, making it a safer option, particularly for immunocompromised individuals. Although it may induce a weaker immune response compared to ACAM2000, JYNNEOS avoids the skin reactions at the vaccination site commonly associated with ACAM2000 [16]. Nevertheless, this weaker immune response means that JYNNEOS offers less protection than the more potent but potentially riskier ACAM2000 [65]. On the other hand, LC16m8, derived from a highly attenuated cowpox virus strain, strikes a balance between safety and immunogenicity. It reduces side effects while inducing a robust immune response. However, further clinical evidence is needed to fully validate its efficacy [73] (Table 2). The aforementioned three smallpox vaccines are crucial in fighting the ongoing mpox outbreak. Their effectiveness in MPXV control and prevention is not entirely adequate, and their side effects hinder their protective function in specific demographics [8]. Therefore, in response to the current mpox outbreak, there is an urgent need for a safer, more effective vaccine specifically designed to target MPXV.

This research goal was successfully achieved through mRNA and recombinant protein vaccines. mRNA vaccines have attracted attention in research due to their fast development, robust response to viral mutations, and simple mass production. The research has indicated that mRNA vaccines and recombinant protein vaccines possess excellent immunogenicity and provide protection in animal models [75]. For instance, the mRNA-1769 vaccine can significantly decrease lesions and viral replication while eliciting a robust immune response. The DAM recombinant protein vaccine, developed by the Institute of Microbiology at the Chinese Academy of Sciences, has also demonstrated notable protective effects and robust neutralization capabilities. Compared with existing vaccines, the new vaccines show significant advantages in multiple aspects. In terms of immunogenicity, mRNA vaccines elicit a stronger immune response than MVA-BN and match or even surpass ACAM2000 in effectiveness [65]. The recombinant protein vaccine DAM offers a protective efficacy comparable to traditional live-attenuated vaccines but with enhanced safety [70]. In contrast, JYNNEOS has relatively weaker immunogenicity. Regarding safety, the new vaccines represent a marked improvement. While ACAM2000 carries a high risk of side effects, mRNA and recombinant protein vaccines are associated with milder adverse reactions and are safer for immunocompromised individuals. In terms of their adaptability, the new vaccines have a broader application scope, which better satisfies the demand for diverse vaccination needs and provides more robust protection across different populations [76]. The ongoing mutations of the MPXV present a challenge for vaccine development, necessitating an exploration of its pathogenic mechanisms and mutation traits to advance more effective, long-lasting, and safe vaccines. Moving forward, large-scale randomized clinical trials will be necessary to assess the safety and effectiveness of the new mpox vaccines.

Current treatment strategies for mpox encompass antiviral drugs, supportive care, and immunomodulatory therapy. Tecovirimat, an FDA-approved antiviral drug for the treatment of mpox virus infection, is effective in alleviating patients’ symptoms and shortening the disease course [77]. It is particularly suitable for severely ill patients and those who are immunocompromised [12]. Cidofovir, which interferes with viral DNA replication by inhibiting the viral DNA polymerase, has a more restricted application due to its nephrotoxicity [78,79]. Brincidofovir, a prodrug of Cidofovir, boasts better oral bioavailability and lower nephrotoxicity. It demonstrated effectiveness against MPXV in animal models [80]. Supportive therapy centers on providing symptomatic treatment and wound care. For instance, non-steroidal anti-inflammatory drugs can be employed to relieve symptoms such as pain and fever. It is also crucial to keep the wound clean and dry to prevent secondary infection [81]. Among the available immunomodulatory therapies, immunoglobulin therapy can be administered to severely immunocompromised patients. However, anti-inflammatory treatments should be used cautiously to avoid undermining the immune system’s ability to combat the virus. To explore combination therapy options, researchers are investigating the simultaneous use of multiple antiviral drugs with different mechanisms of action, such as combining Tecovirimat with PA104, with the aim of enhancing treatment efficacy and reducing the likelihood of viral resistance [19]. Cutting-edge therapeutic approaches, including gene therapy and immunomodulatory therapies, also create new hope for mpox treatment [82]. Drug repurposing studies have identified several drugs with potential antiviral activity by screening existing FDA-approved drugs, like doxycycline, niclosamide, sunitinib, oestradiol, and medroxyprogesterone acetate [83]. Monoclonal antibody studies are focused on developing monoclonal antibodies against specific antigens of the MPXV to specifically neutralize it [19]. Additionally, research into traditional and alternative therapies has identified certain plant extracts and natural compounds with antiviral and anti-inflammatory properties, which may provide new ideas for future treatments [82].

An effective response to the global mpox outbreak should involve more than just vaccination or treatment strategies, and consider socioeconomic factors such as poverty, healthcare access, and urbanization [84]. Poverty drives people deeper into forests in search of food and resources, increasing exposure to wildlife—the natural host of the MPXV. Hunting, handling, and consuming wildlife further heighten infection risks. Moreover, poverty restricts access to healthcare, delaying effective treatment and accelerating viral spread [85]. Rapid urbanization and population growth have created crowded living conditions, facilitating human-to-human transmission. In urban areas with poor sanitation and limited medical resources, the virus spreads more rapidly [86]. Furthermore, urbanization drives large-scale population movements, accelerating regional virus spread and triggering transnational public health crises. The interplay of poverty and urbanization intensifies the severity of the mpox epidemic [84,87]. To effectively address mpox, these factors must be considered in the development of targeted public health strategies and interventions. These include improving the economic conditions of poor areas, increasing employment opportunities, and reducing reliance on wildlife; enhancing urban public health infrastructure to improve healthcare access and quality; and strengthening mpox surveillance and prevention, particularly in high-risk, impoverished, and rapidly urbanizing areas [88]. In addition to other problems, the fair distribution of vaccines globally is a pressing issue that needs resolution.

## 6. Conclusions

Given the current progress of vaccine research, both domestically and internationally, the immunity now provided via the smallpox vaccine is the only protection against MPXV. Second- and third-generation vaccines, which are safer and have better immune efficacy, have replaced first-generation smallpox vaccines [47]. Additionally, clinical trials provided evidence that smallpox vaccines, such as MVA-BN, caused no serious adverse reactions in immunocompromised subjects. This finding offers a key foundation for expanding the target population for vaccination and eliminating mortality due to mpox infection [89]. MPXV Clade Ib remains widespread and continues to spread globally due to mutations that increase the pathogenicity and transmissibility of MPXV. To address this challenge, an MPXV-specific vaccine should be developed with the ability to rapidly adapt to emerging strains, ensuring effective targeting of evolving MPXV variants. Key objectives include improving vaccine safety and efficacy while reducing viral pathogenicity and transmissibility. The developed vaccines are more suitable for those with immune system disorders, skin problems, or cardiovascular problems [50]. The mRNA-1769 vaccine is currently being evaluated in an ongoing Phase I/II clinical trial (NCT05995275) to assess the safety and immunogenicity of an mRNA-based *Orthopoxvirus* vaccine [65]. BNT166 is also in a phase 1/2 clinical trial to determine its safety and immunogenicity [66]. At the pilot stage, DAM was developed by Wang et al. in collaboration with Shanghai Junshi Biological Products, which developed a one-step purification of industrially produced cell lines and DAM with a yield of 1.18 g/L. This innovation offers a safer and easier-to-scale vaccine option for the prevention and control of MPXV [70]. Overall, new advances in vaccines are boosting confidence in the next round of vaccines to prevent and treat mpox. Despite this progress, there are still some gaps in vaccine distribution, the accessibility of therapeutic drugs, and regional equity. In addition, the surveillance, diagnosis, treatment, and management of mpox presents challenges in some or varied countries or regions. Although PCR is the most commonly used MPXV diagnostic because of its high sensitivity and specificity, it cannot be used in point-of-care testing (POCT) due to the equipment and environmental demands [90]. Within this context, CRISPR/Cas technology is emerging as a new means of nucleic acid detection with no requirement for dedicated laboratory equipment and a controlled environment; this is expected to serve as a POCT candidate. This technology can quickly and precisely test mpox nucleic acids in various countries and regions [91]. Therefore, a better understanding of mpox and an improvement in the entire prevention system could provide valuable insights for dealing with severe cases and implementing effective scientific prevention and control.

## Figures and Tables

**Figure 1 vaccines-13-00466-f001:**
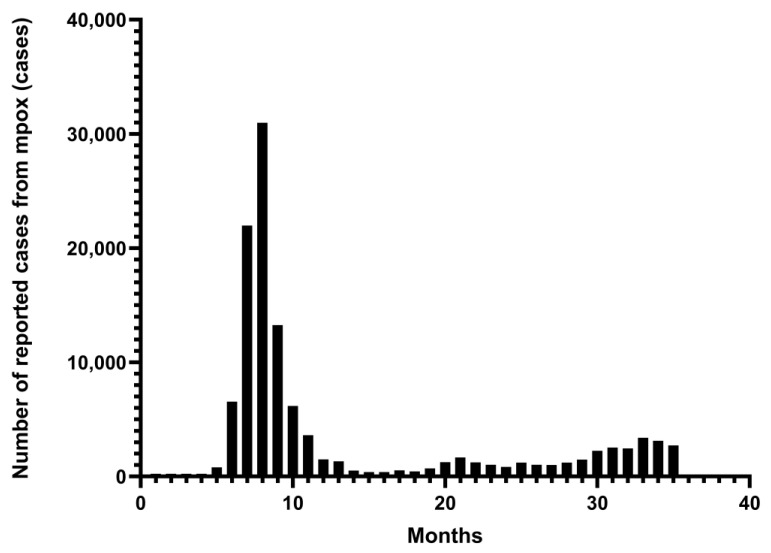
From January 2022 to November 2024. Monthly number of mpox cases reported in 128 Member States across all six WHO regions. The data are from the WHO Global Aggregate of Mpox Cases. This figure was drawn by Graphpad Pism 9.5.

**Figure 2 vaccines-13-00466-f002:**
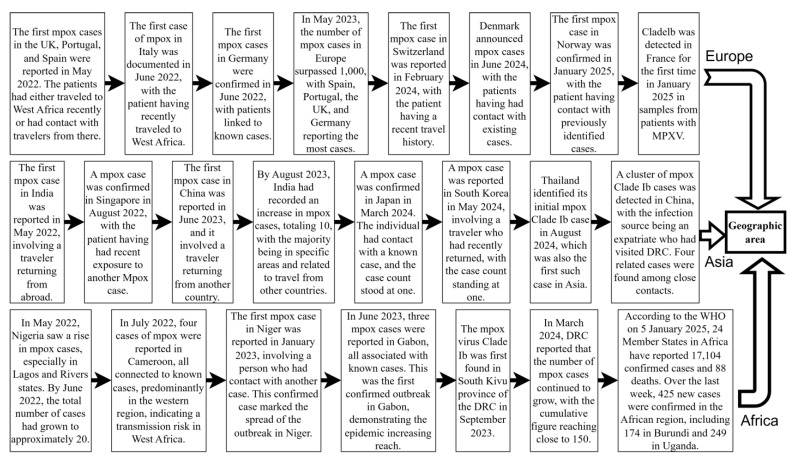
Global mpox outbreaks and related important events since 2022. This figure was drawn using Figdraw2.0.

**Figure 3 vaccines-13-00466-f003:**
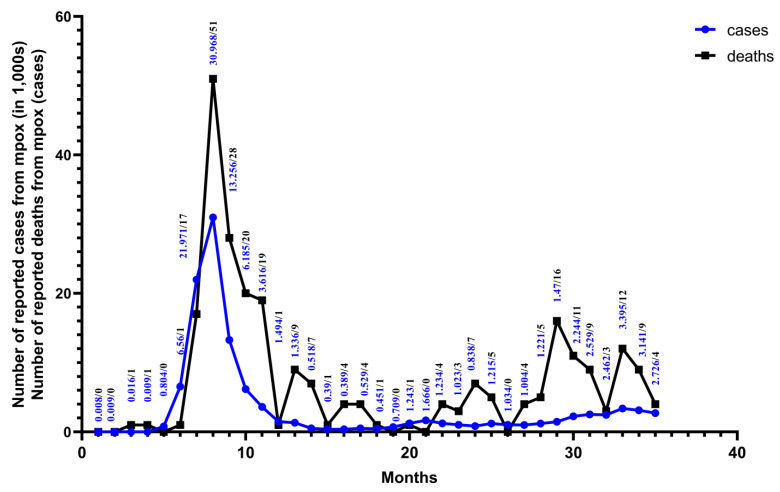
From January 2022 to November 2024. Monthly numbers of cases and deaths from mpox are reported for 128 Member States across all six WHO regions. The data are from the WHO Global Aggregate of Mpox Cases. This figure was drawn using Graphpad Pism.

**Table 1 vaccines-13-00466-t001:** Comparison of similarities and differences between four mpox clades.

Clade	Historical Period	Geographic Distribution	Transmission Dynamics	The Main Affected Population	Mortality Rates	Mutation Rates	Trends	Public Health Impact	References
Clade Ia	1970–2024	Central Africa (mainly central and western DRC)	Zoonotic (more than 70%) with small human-to-human transmission	Mostly children	Higher virulence, approximate 1–11% mortality rate	Stable genome, low mutation rate	The incidence was low before 2010, and then increased year by year.	Prevalent mainly in traditional areas of Africa (Central Africa).	[14]
Clade Ib	2023–2025	A global epidemic	Human-to-human transmission (household transmission, vertical transmission, contact transmission)	Children, pregnant women, sex workers	Highly virulent, higher mortality than Clade II	Very unstable genomes, extremely high mutation rate	The first case occurred in 2023. Widespread outbreak and spread globally in 2024	On 14 August 2024, mpox was declared a PHEIC by the WHO again.	[4,22]
Clade IIa	1970–2018	Mainly in West Africa	Zoonotic with Widespread human-to-human transmission	Mostly adults	Low virulence, <1% mortality	Moderately stable genome, low mutation rate	1970–2018 outbreak and transmission in West Africa.	Prevalent mainly in traditional areas of Africa (West Africa), with low frequency of outbreaks and low mortality rates.	[15]
Clade IIb	2022–2023	A global epidemic	Human-to-human transmission (contaminant transmission, sexual transmission, contact transmission)	Adults (men who have sex with men)	Lowest toxicity, <1% mortality	Unstable genome, high mutation rate	Outbreak in Nigeria in 2017. Widespread global outbreak and transmission in 2022.	On 23 July 2022, mpox was declared a PHEIC by the WHO.	[9]

Table 1 compares the historical period, geographic distribution, transmission dynamics, mortality rates, mutation rates, trends, and public health impact of each clade.

**Table 2 vaccines-13-00466-t002:** Similarities and differences between ACAM2000, JYNNEOS, and LC16m8 vaccines.

Vaccine	Type	Effectiveness	Safety	Side Effect	Applicable Population	Not Applicable Population	Routes of Administration	Storage Method	References
ACAM2000 (developed by Emergent BioSolutions; produced by Sanofi Pasteur)	Replicative, live attenuated vaccine based on cowpox virus.	Immunogenicity comparable to first-generation vaccine Dryvax. The length of immunity can differ among people, with research indicating that it might last for a minimum of 2 years.	Higher than Dryvax, although serious side effects may still occur. It is possible to administer via vaccine like the flu, but a doctor should be consulted to determine the most suitable choice.	Injection site pain, fever, flu-like symptoms, severe allergic reactions, encephalitis, progressive cowpox, myocarditis, and/or pericarditis.	Non-pregnant, immunocompetent individuals. Recommended for individuals at high risk of transmission.	Individuals with impaired cardiac function, pregnant women, newborns, and individuals with impaired immune function (HIV)	One dose via a percutaneous route. No definitive official guidance mandates a booster vaccination.	General refrigeration	[45]
JYNNEOS (developed by Bavarian Nordic; produced by Bavarian Nordic)	Non-replicating, live vaccine based on modified Ankara poxvirus.	Slightly lower immunogenicity than ACAM2000. The length of immunity can differ among people, with research indicating that it might last for a minimum of 2 years.	Much safer than ACAM2000, with less severe and manageable side effects. It is possible to administer via a vaccine like the flu, but a doctor should be consulted to determine the most suitable choice.	Minor injection site reactions and transient systemic symptoms, pain, fatigue, itching, chills, and nausea.	Recommended for individuals at high risk of transmission. Pre-exposure prophylaxis can be administered to HIV patients, but requires evaluation on a case-by-case basis.	Individuals with a low risk of transmission.	Two doses inoculated via SC. Individuals need a booster shot every 2–10 years if they remain exposed to MPXV.	General refrigeration	[53,55]
LC16m8 (developed by Tokyo University of Science; produced by The Research Foundation for Microbial Diseases of Osaka University in Japan)	Minimally replicative, attenuated live vaccine based on the cowpox virus.	Japanese Clinical Trial Shows Good Immunogenicity. While data are limited, it is expected to be similar to other vaccines of the same type.	Japanese clinical trials show good safety. It is possible to administer via a vaccine like the flu, but a doctor should be consulted to determine the most suitable choice.	Limited detailed clinical data, but generally considered to have fewer side effects.	Recommended for individuals at high risk of transmission.	Contraindicated in individuals with severe immunosuppression (cancer patients and HIV/AIDS patients receiving certain treatments) and severe allergies.	A single-dose scratch vaccination using a specially designed fork needle. No definitive official guidance mandates a booster vaccination.	General refrigeration	[61,74]

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
