# Peer review of "Recent Advances in Mpox Epidemic: Global Features and Vaccine Prevention Research"

_vaccines, 2025, doi:10.3390/vaccines13050466_

Round 1
Reviewer 1 Report
Comments and Suggestions for Authors
The review is well written, however there are a fewpoints that might be improved. Firstly the authors refer to Mpox virulence but the figure 3 provides only information on mortality and not on case-falatity. I would prefer to have only a figure on cases and deaths, and then to report in the text information about case-fatality of the dfferent clades. Secondly, the Discussion is mainly focused on vaccine research and development, thus instead of a distinct chapter it could be just a paragraph of the chppater oon vaccines.
Author Response
Dear Esteemed Reviewer#1,
Thank you for your invaluable feedback and insightful suggestions. All of the changes in our manuscript are highlighted in red and uploaded in Supplemental Files. The revised edition was uploaded in Main Manuscript. We greatly appreciate your expertise and have thoroughly considered each point raised in your comments. Please find below a detailed response addressing each of your concerns:
- Firstly, the authors refer to Mpox virulence but the figure 3 provides only information on mortality and not on case-falatity. I would prefer to have only a figure on cases and deaths, and then to report in the text information about case-fatality of the dfferent clades.
Response: Thank you very much for your comments, we have updated Figure3 with figures on cases and deaths. In addition, we have reported information on the morbidity and mortality rates for the different branches in the text (Lines 77 ~80) and Table 1.
- Secondly, the Discussion is mainly focused on vaccine research and development, thus instead of a distinct chapter it could be just a paragraph of the chppater oon vaccines.
Response: Thank you very much for your comments. We have included vaccine research and development (4.3) as an independent subsection. Furthermore, we add a little to the discussion. (Lines 322~394)
Once again, we express our sincere appreciation for your expert evaluation and thoughtful recommendations. We have taken great care to address each of your concerns and have provided detailed explanations and revisions to ensure the accuracy and scientific rigor of the manuscript.
Thank you for your time and continued support.
Best regards,
Yours sincerely,
Xiaoping Li
Email: li-xp@zjsru.edu.cn
Reviewer 2 Report
Comments and Suggestions for Authors
The article provided some interesting data on the advances in the mpox vaccine, however there is a lack of several information’s about the three vaccines that are described in the manuscript:
- Vaccine manufactures?
- To whom is recommended?
- How many doses and boosters?
- Duration of immunity (durability of protection)?
- Coadministration with other vaccines?
- Recommendations for people living with HIV?
- Effectivity of the vaccines?
- Routes of administration (any difference on side effects?)
- Side effects?
- Get mpox even after been fully vaccinated?
- Next generation of mpox vaccines?
Minor corrections:
- Please use mpox instead Mpox
- Use numerals to express numbers 10 and above, and use words for numbers zero through nine
- Avoid beginning a sentence with an acronym or an abbreviation
Author Response
Dear Esteemed Reviewer#2,
Thank you for your invaluable feedback and insightful suggestions. All of the changes in our manuscript are highlighted in red and uploaded in Supplemental Files. The revised edition was uploaded in Main Manuscript. We greatly appreciate your expertise and have thoroughly considered each point raised in your comments. Please find below a detailed response addressing each of your concerns:
- The article provided some interesting data on the advances in the mpox vaccine, however there is a lack of several information’s about the three vaccines that are described in the manuscript:
Vaccine manufactures?
To whom is recommended?
How many doses and boosters?
Duration of immunity (durability of protection)?
Coadministration with other vaccines?
Recommendations for people living with HIV?
Effectivity of the vaccines?
Routes of administration (any difference on side effects?)
Side effects?
Get mpox even after been fully vaccinated?
Next generation of mpox vaccines?
Response: Thank you very much for your comments. In the section on recent advances in vaccine development (4.3), we shared insights on the next generation of monkeypox vaccines and confirmed that mpox infection can still occur even with full vaccination. The following questions were also answered in Table 2.
Vaccine manufactures?
To whom is recommended?
How many doses and boosters?
Duration of immunity (durability of protection)?
Coadministration with other vaccines?
Recommendations for people living with HIV?
Effectivity of the vaccines?
Routes of administration (any difference on side effects?)
Side effects?
- Please use mpox instead Mpox
Response: Thank you very much for your comments, we have corrected our mistakes throughout the manuscript!
- Use numerals to express numbers 10 and above, and use words for numbers zero through nine
Response: Thank you very much for your comments, we have corrected our mistakes throughout the manuscript!
- Avoid beginning a sentence with an acronym or an abbreviation
Response: Thank you very much for your comments, we have corrected our mistakes throughout the manuscript. (Line 130)
Once again, we express our sincere appreciation for your expert evaluation and thoughtful recommendations. We have taken great care to address each of your concerns and have provided detailed explanations and revisions to ensure the accuracy and scientific rigor of the manuscript.
Thank you for your time and continued support.
Best regards,
Yours sincerely,
Xiaoping Li
Email: li-xp@zjsru.edu.cn
Reviewer 3 Report
Comments and Suggestions for Authors
The manuscript titled "Recent Advances in Mpox Epidemic: Global Features and Vaccine Prevention Research" provides a comprehensive review of the Mpox (monkeypox) virus, its epidemiology, and advancements in vaccine development.
The manuscript covers a wide range of topics related to Mpox, including its etiology, global features, transmission dynamics, and vaccine prevention strategies. This holistic approach ensures that readers gain a thorough understanding of the current state of Mpox research. The authors have included recent studies and developments up to 2025, providing the latest insights into the Mpox epidemic. This timeliness is crucial for understanding the rapidly evolving situation with Mpox. Given the global concern about emerging infectious diseases, this review is highly relevant. It addresses the need for awareness and vaccination efforts, especially among high-risk populations, which is critical for public health planning and response. The manuscript includes an extensive list of references, supporting the claims and discussions with credible scientific literature. This strengthens the validity and reliability of the information presented. The paper is well-structured, with clear sections that logically progress from introduction to conclusion. This organization facilitates readability and comprehension. The review not only summarizes existing knowledge but also highlights practical implications for managing Mpox outbreaks. It emphasizes the importance of vaccination and offers insights into effective scientific prevention and control strategies.
While the manuscript compiles a significant amount of data and findings, it lacks a critical analysis of the studies cited. A deeper critique of the methodologies, limitations, and potential biases in the referenced research would enhance the depth of the review. Although the manuscript touches on the global spread of Mpox, it does not sufficiently address socioeconomic factors that could influence the transmission and management of the disease. Including such discussions could provide a more nuanced understanding of the epidemic's dynamics. The review primarily focuses on vaccines and preventive measures but gives limited attention to treatment options for Mpox. Discussing current and potential treatments would offer a more complete overview of the disease management landscape. Some sections generalize data without providing specific details or context. For example, while summarizing vaccine effectiveness, more detailed comparisons between different vaccines and their efficacies in various populations would be beneficial. There are areas where content seems repetitive, particularly in the sections discussing vaccine development and characteristics. Streamlining these sections could improve the manuscript’s conciseness and clarity.
Integrate a critical evaluation of the methodologies, results, and conclusions of the cited studies. Highlight any discrepancies or gaps in the research to provide a balanced perspective. Include a discussion on how socioeconomic factors, such as poverty, healthcare access, and urbanization, impact the spread and management of Mpox. This would offer a more comprehensive view of the epidemic’s challenges. Add a section dedicated to current treatment strategies and ongoing research into new therapies for Mpox. This would complement the focus on vaccines and prevention, offering a more rounded approach to disease management. When discussing vaccines, include detailed comparisons of their efficacy, safety profiles, and implementation challenges in different settings. This would help readers understand the nuances of each vaccine option. Review the manuscript to identify and remove any redundant information, ensuring that each section adds unique value to the overall narrative.
Overall, the manuscript provides a valuable and timely review of the Mpox epidemic and advances in vaccine prevention. It successfully consolidates a vast array of information into a coherent narrative that underscores the importance of vigilance and preparedness in managing emerging infectious diseases. Addressing the identified weaknesses and incorporating the suggested improvements will further enhance the quality and impact of the review. By doing so, the authors can ensure that their work serves as a robust reference for researchers, public health officials, and policymakers involved in combating the Mpox epidemic.
Author Response
Dear Esteemed Reviewer#3,
Thank you for your invaluable feedback and insightful suggestions. All of the changes in our manuscript are highlighted in red and uploaded in Supplemental Files. The revised edition was uploaded in Main Manuscript. We greatly appreciate your expertise and have thoroughly considered each point raised in your comments. Please find below a detailed response addressing each of your concerns:
- While the manuscript compiles a significant amount of data and findings, it lacks a critical analysis of the studies cited. A deeper critique of the methodologies, limitations, and potential biases in the referenced research would enhance the depth of the review.
Lack of critical analysis: The methodological limitations of the cited studies were not
assessed in depth, weakening the depth of the review.
Response: Thank you very much for your comments. We have included advances in the mpox vaccines (in part 4 of the article) as an independent section. Additionally, we have expanded the discussion section to strengthen the research critique. This expansion highlights research differences and shortcomings by comparing the new generation of mpox vaccines with the three currently mainstream vaccines. (Lines 245~288)
- Integrate a critical evaluation of the methodologies, results, and conclusions of the cited studies. Highlight any discrepancies or gaps in the research to provide a balanced perspective. Include a discussion on how socioeconomic factors, such as poverty, healthcare access, and urbanization, impact the spread and management of Mpox.
Lack of socio-economic factors: the impact of poverty and healthcare accessibility on monkeypox transmission was not explored, resulting in an incomplete analysis.
Response: Thank you very much for your comments. We have added a socio-economic analysis to the discussion to explore the influence of poverty and urbanization on monkeypox prevalence. (Lines 375~394)
- This would offer a more comprehensive view of the epidemic’s challenges. Add a section dedicated to current treatment strategies and ongoing research into new therapies for Mpox. This would complement the focus on vaccines and prevention, offering a more rounded approach to disease management.
Insufficient coverage of treatment strategies: limited discussion of monkeypox treatment options failed to develop a complete perspective on disease management.
Response: Thank you very much for your comments. We have supplemented with current treatment options and disease management programmes in the discussion section. (Lines 346~374)
- When discussing vaccines, include detailed comparisons of their efficacy, safety profiles, and implementation challenges in different settings. This would help readers understand the nuances of each vaccine option.
Insufficient data details: the comparison of vaccine effectiveness lacks analysis of differences in specific populations, and some of the conclusions appear to be general.
Response: Thank you very much for your comments. In part 5 of the article (5. Discussion) and Table 2, we have enhanced the vaccine comparisons by including evaluations of efficacy, safety, and implementation challenges across various populations. Additionally, we have enhanced and incorporated the following points to more accurately convey the details of the comparisons.
Vaccine manufactures?
To whom is recommended?
How many doses and boosters?
Duration of immunity (durability of protection)?
Coadministration with other vaccines?
Recommendations for people living with HIV?
Effectivity of the vaccines?
Routes of administration (any difference on side effects?)
Side effects?
Get mpox even after been fully vaccinated?
Next generation of mpox vaccines?
- Review the manuscript to identify and remove any redundant information, ensuring that each section adds unique value to the overall narrative.
Repetition of content: redundant information in the section on vaccine development, affecting reading fluency.
Response: Thank you very much for your comments. We have streamlined redundancies, cutting out repetitive expressions in the text (Lines 38~40 55~57 77~80 and so on) and enhancing the cohesiveness of the text.
Once again, we express our sincere appreciation for your expert evaluation and thoughtful recommendations. We have taken great care to address each of your concerns and have provided detailed explanations and revisions to ensure the accuracy and scientific rigor of the manuscript.
Thank you for your time and continued support.
Best regards,
Yours sincerely,
Xiaoping Li
Email: li-xp@zjsru.edu.cn
Reviewer 4 Report
Comments and Suggestions for Authors
I found the review to be thorough and well-researched.
I have a few minor changes for the authors:
- line 89: change text to read "Human infection with Clade IIb has an incubation period of 7 to 14 days..."
- line 95: change text to read "Human infection with Clade Ib has an incubation period of 5 to 13 days..."
- line 103: the statement "MPXV is well tolerated" in this section is confusing and seems out of place - please explain what is meant by this or remove this text
- line 104: remove the word "good" as an adjective describing the word vectors
- line 104: rather than say "for a long time," which is rather vague, could the authors provide some detail with a range of known times that the virus can dwell on a fomite and still remain infectious?
- line 174: change "maybe" to "may be"
- line 183: the section heading should read "Advances in Vaccines against Mpox" ["vaccine" must be made plural as there isn't just one option]
Author Response
Dear Esteemed Reviewer#4,
Thank you for your invaluable feedback and insightful suggestions. All of the changes in our manuscript are highlighted in red and uploaded in Supplemental Files. The revised edition was uploaded in Main Manuscript. We greatly appreciate your expertise and have thoroughly considered each point raised in your comments. Please find below a detailed response addressing each of your concerns:
- line 89: change text to read "Human infection with Clade IIb has an incubation period of 7 to 14 days..."
Response: Thank you very much for your comments, we have corrected it. (Lines 84~85)
- line 95: change text to read "Human infection with Clade Ib has an incubation period of 5 to 13 days..."
Response: Thank you very much for your comments, we have corrected it. (Lines 89~90)
- line 103: the statement "MPXV is well tolerated" in this section is confusing and seems out of place - please explain what is meant by this or remove this text
line 104: remove the word "good" as an adjective describing the word vectors
line 104: rather than say "for a long time," which is rather vague, could the authors provide some detail with a range of known times that the virus can dwell on a fomite and still remain infectious?
Response: Thank you very much for your comments. Due to the relevance of these three points, we have adjusted the entire sentence. Change the sentence to "The stability of MPXV enables it to survive for a certain amount of time in vitro, using contaminated surfaces or environments as a medium. It has a half-life of up to 38.75 days in dried blood, 4.57 days in dried semen, and 5.74 days in wastewater." We describe MPXV in terms of stability and specifically provide a known timeframe in which MPXV can remain on contaminants and still be infectious for a known time frame. (Lines 98 ~101)
- line 174: change "maybe" to "may be"
Response: Thank you very much for your comments, we have corrected it. (Line 163)
- line 183: the section heading should read "Advances in Vaccines against Mpox" ["vaccine" must be made plural as there isn't just one option]
Response: Thank you very much for your comments, we have corrected it throughout the manuscript. (Line 172)
Once again, we express our sincere appreciation for your expert evaluation and thoughtful recommendations. We have taken great care to address each of your concerns and have provided detailed explanations and revisions to ensure the accuracy and scientific rigor of the manuscript.
Thank you for your time and continued support.
Best regards,
Yours sincerely,
Xiaoping Li
Email: li-xp@zjsru.edu.cn